# LLM4Causal: Democratized Causal Tools for Everyone via Large Language Model

**Haitao Jiang, Lin Ge, Yuhe Gao,**[*] **Jianian Wang**[*] **& Rui Song**[†]
Department of Statistics
North Carolina State University
Raleigh, NC 27607, USA
{hjiang24,lge,ygao32,jwang95,rsong}@ncsu.edu

## Abstract

Large Language Models (LLMs) have shown their success in language understanding and reasoning on general topics. However, their capability to perform inference based on user-specified structured data and knowledge in corpus-rare concepts, such as causal decision-making is still limited. In this work, we explore the possibility of fine-tuning an open-sourced LLM into LLM4Causal, which can identify the causal task, execute a corresponding function, and interpret its numerical results based on users' queries and the provided dataset. Meanwhile, we propose a data generation process for more controllable GPT prompting and present two instruction-tuning datasets: (1) Causal-Retrieval-Bench for causal problem identification and input parameter extraction for causal function calling and (2) Causal-Interpret-Bench for in-context causal interpretation. By conducting end-to-end evaluations and two ablation studies, we showed that LLM4Causal can deliver end-to-end solutions for causal problems and provide easy-to-understand answers, which significantly outperforms the baselines.

## 1 Introduction

Recently, Transformer-based LLMs containing billions of parameters are gaining popularity and are widely applied in fields such as education, legal services, and medicine (Kasneci et al., 2023; Chen et al., 2021). LLMs such as GPT-3 (Brown et al., 2020), GPT-4, and LLaMA (Touvron et al., 2023a) have shown impressive performance in multiple natural language processing tasks such as question-answering, common-sense reasoning, and translation (Zhao et al., 2023). These superior performances of LLMs have also motivated explorations on LLMs' applications on the causal decision-making (Song et al., 2023) procedures, which include various crucial tasks in real-life such as causal structure learning (CSL) tasks (Spirtes et al., 2000; Glymour et al., 2019), causal effect learning (CEL) tasks (Yao et al., 2021; Hicks & Tingley, 2011) and causal policy learning (CPL) tasks (Chakraborty & Murphy, 2014; Sutton & Barto, 2018; Zeng et al., 2023).

Many recent works that applied LLM in causal decision-making tasks are focused on CSL, where LLM's internal knowledge gained during its training process is exploited in learning the causal relations among variables. For instance, both Kıcıman et al. (2023) and Long et al. (2023) propose to obtain the causal relationships between variables by directly querying GPT models using the variable names, assuming that LLMs can extract internal causal knowledge of those variables from the large corpus it was trained on. In a further development by Ban et al. (2023), LLM's internal causal knowledge is utilized to guide the optimization process of score-based methods as soft or hard constraints, enhancing the performance compared to using the traditional CSL methods alone. However, this line of work heavily relies on the quality of the knowledge in LLM's training corpus and lacks step-by-step reasoning or

---

[*]equal contribution
[†]This work is not related to the author's position at Amazon.

interpretation. A natural choice to overcome these limitations is to ask the LLM to complete the causal tasks utilizing existing causal decision-making tools (such as CausalML(Chen et al., 2020), CausalDM (Song et al., 2023), causal-learn (Zheng et al., 2023)) and input data uploaded by users. In literature, Code Interpreter (OpenAI, 2023a), ToolLLM (Qin et al., 2023b), and GPT4Tools(Yang et al., 2023) have leveraged the function calling feature of LLM in general data analysis tasks for user input datasets. However, directly applying those methods to causal-related tasks and datasets may lead to several issues: first, since these models are not tailored for causal tasks, they frequently hallucinate and mislead the user with irrelevant contexts, such as providing correlation analysis results when causal effect analysis is expected, as shown in Figure 1a; second, most of these methods fail to provide end-to-end result delivery (see Figure 1b) and easy-to-understand interpretation for causal-related tasks; third, these methods lack information about newly released approaches that are not included in LLM's training corpus (see Figure 1c).

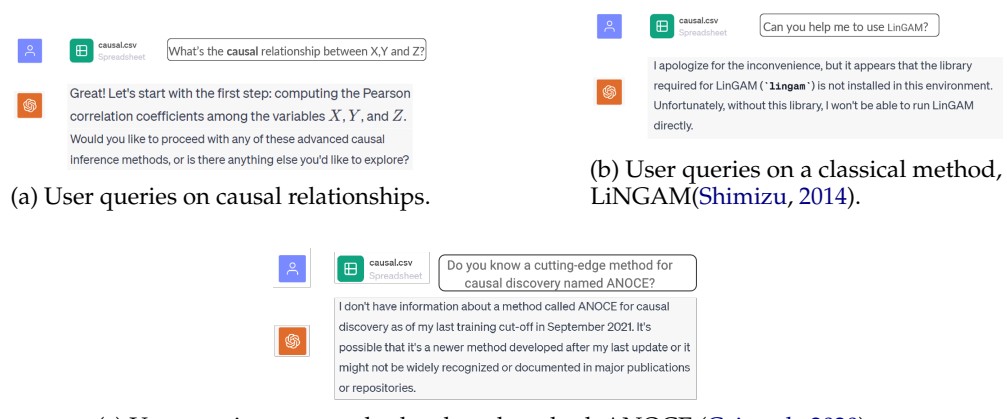

(a) User queries on causal relationships.

(b) User queries on a classical method, LiNGAM(Shimizu, 2014).

(c) User queries on a newly-developed method, ANOCE (Cai et al., 2020).

Figure 1: User interaction with ChatGPT on causal related questions.[1]

To overcome these challenges, in this study, we develop an end-to-end user-friendly large language model with causal decision-making ability for general audiences. As illustrated in Figure 2, the proposed model consists of three major steps: user request interpretation, causal tools assignment and execution, and output interpretation. Upon receiving a user query and an uploaded data file, the initial step identifies the pertinent causal task and extracts query details—including dataset name, task type, and variable of interest, among other variables—into a structured JSON summary. The LLM4Causal model gains the ability to convert natural language user queries into JSON summaries by fine-tuning a pre-trained LLM on the Causal-Retrieval-Bench dataset. A well-designed data generation pipeline is proposed to ensure the quality of Causal-Retrieval-Bench, in terms of both data variety and accuracy. In the second step, the system automatically chooses the causal learning algorithm according to the task type detailed in the structured JSON data, executes the selected algorithm to analyze the dataset, and collects the algorithm's output. The outputs are then translated into easily understandable language in the third step, using the LLM4Causal model, which has been further fine-tuned with the Causal-Interpret-Bench dataset to generate high-quality interpretations.

Our main contributions could be summarized as follows:

- This paper is the first to construct an end-to-end user-friendly large language model (**LLM4Causal**) with causal decision-making ability. **LLM4Causal** could be easily used for general audiences, which addresses the weakness of the current LLM applications on causal tasks. It has the capability of i) interpreting user requests by causal task classification and information extraction, ii) assigning causal tools and executing the corresponding algorithm, and iii) providing an easy-to-understand interpretation of the algorithm output.

---

[1]The chat histories could be found in link 1, link 2, and link 3.

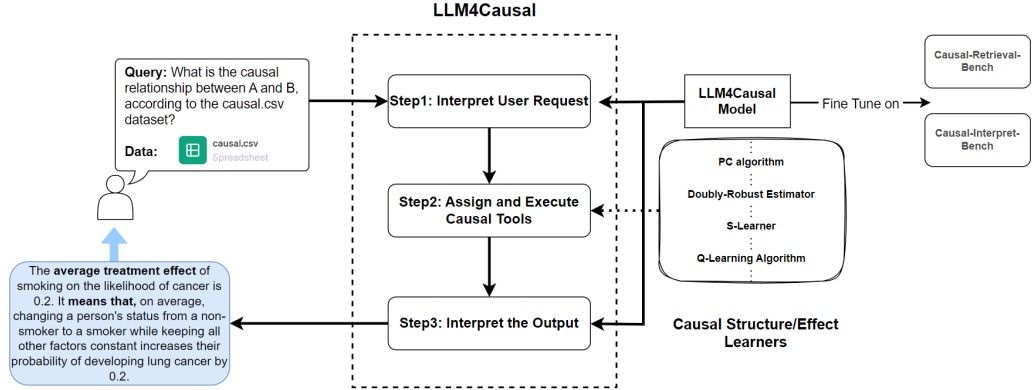

Figure 2: A flowchart of the LLM4Causal consists of three major steps: user request interpretation, causal tools assignment and execution, and output interpretation.

- We have proposed a three-step data generation pipeline that combines LLM text generation and human annotation to create fine-tuning datasets. With this well-designed pipeline, we have collected two high-quality benchmark datasets, **Causal-Retrieval-Bench** for causal function calling and **Causal-Interpret-Bench** for causal interpretation, with outstanding data variety and accuracy.
- The proposed **LLM4Causal** model was extensively evaluated in three major causal decision-making tasks: Causal effect estimation, Causal structure discovery, and Causal policy learning. The proposed method has shown superior performance compared to the benchmark methods.

## 2 Problem formulation

In this paper, we aim to augment existing pre-trained LLMs with the proficiency to address causal inquiries, thereby opening the door to causal decision-making processes for general audiences. As introduced previously, causal decision-making tasks can be classified into three categories: 1) causal structure learning (CSL), 2) causal effect learning (CEL), and 3) causal policy learning (CPL). Each of these primary categories consists of multiple tasks, and our study focuses on five key tasks that are particularly relevant to the interests of our target audience. Let us denote the user query as $\mathcal{Q}$, the dataset to be analyzed as $\mathcal{D}$, the node set whose interrelationship is of interest as $\mathcal{X}$, the treatment variable as $\mathcal{A}$, the response variable as $\mathcal{Y}$, the mediator variables as $\mathcal{M}$, and the condition of the subpopulation of interest as $\mathcal{S}$. The five causal tasks are summarized in Table 1.

| Task | Dataset $(\mathcal{D})$ | Nodes $(\mathcal{X})$ | Treatment $(\mathcal{A})$ | Response $(\mathcal{Y})$ | Mediator $(\mathcal{M})$ | Condition $(\mathcal{S})$ | Function Output Format |
|------|---------|-------|-----------|----------|----------|-----------|------------------------|
| CGL | ✓ | ✓ | | | | | Causal Graph $(\mathcal{G})$ |
| ATE | ✓ | | ✓ | ✓ | | | Numeric Value |
| HTE | ✓ | | ✓ | ✓ | | ✓ | Numeric Value |
| MA | ✓ | | ✓ | ✓ | ✓ | | Numeric Values |
| OPO | ✓ | | ✓ | ✓ | | ✓ | Treatment Level |

Table 1: Constructions of causal tasks

The CSL category includes **Causal Graph Learning (CGL)** task, which aims to identify causal relationships between variables. Given $\mathcal{Q}$ and $\mathcal{D}$, the goal of CGL is to learn and report a directed acyclic graph $\mathcal{G}$ that encapsulates the entire causal structure among variables in $\mathcal{X}$ specified in $\mathcal{Q}$. **Average Treatment Effect Estimation (ATE)** falls under the CEL category and aims to quantify the average effect size of an intervention across the entire population. Given $\mathcal{D}$, $\mathcal{A}$, and $\mathcal{Y}$ defined in $\mathcal{Q}$, the goal of ATE is to execute appropriate learners to

measure the difference in counterfactual outcomes between the treated and control groups. **Heterogeneous Treatment Effect Estimation (HTE)**, another task within CEL, extends ATE by assessing effect sizes under specific conditions $\mathcal{S}$, providing insights into the variability of treatment effects across different subpopulations. When additional mediators $\mathcal{M}$ are considered to transmit treatment effects to the response, **Mediation Effect Analysis (MA)** within CEL focuses on decomposing the total treatment effect into direct effects, which are solely due to $\mathcal{A}$, and indirect effects, mediated through additional variables $\mathcal{M}$. Lastly, **Off-Policy Optimization (OPO)**, the only one we considered within the CPL category, served as a one-stop shop for decision-makers. Given $\mathcal{D}$, $\mathcal{A}$, $\mathcal{Y}$, and $\mathcal{S}$ explicitly specified in $\mathcal{Q}$, OPO aims to select suitable policy learners to determine the optimal action expected to maximize the outcome $\mathcal{Y}$.

To tackle the aforementioned diverse tasks using a single LLM, we introduce LLM4Causal. By fine-tuning pre-trained LLMs, LLM4Causal is capable of comprehending causal queries, applying appropriate causal tools to analyze the provided tabular dataset, and providing answers by interpreting numerical results in straightforward and fluent language. More technical details of LLM4Causal is discussed in Section 3.

## 3 Proposed method

In this paper, we introduce a novel three-stage framework to empower a pre-trained LLM to address causality-related tasks, as illustrated in Figure 2. Common approaches to calibrate an LLM checkpoint include Retrieval Augmented Generation (RAG), Prompt Engineering (PE), and Fine-Tuning (FT). However, while RAG (Chen et al., 2024; Gao et al., 2023) can introduce relevant external information into the response process, it cannot improve the model's intrinsic understanding of causality or result interpretation capability, rendering it infeasible. Although various PE strategies (Liu et al., 2023; Haviv et al., 2021) have been developed to enhance the performance of LLMs' responses, PE remains a complex and nuanced art that requires iterative and extensive experimentation to refine the prompts. In contrast, having a clear understanding of the expected performance outcomes makes preparing the 'golden' dataset relatively straightforward, thereby making the fine-tuning approach more direct and effective for our case. As a result, we choose to fine-tune an LLM on carefully crafted datasets. Specifically, LLM4Causal is carefully fine-tuned with our proposed benchmark dataset, *Causal-Retrieval-Bench*, enabling it to achieve better performance than other LLMs in classifying causality-related tasks and retrieving relevant information, such as variable names and values. Then, the framework selects appropriate causal analysis tools based on task classification and retrieved information to produce quantitative results. Finally, further fine-tuned on our proposed *Causal-Interpret-Bench* dataset, LLM4Causal is capable of translating the direct results from functional calls into clear and easily understandable natural language interpretations.

### 3.1 Step1. interpret user request

The purpose of the first step is to translate user questions through two integrated substeps: i) causal task classification and ii) attribute extraction. The first sub-step categorizes the input question into one of the five supported causal tasks (CGL, ATE, HTE, MA, or OPO) by inferring the underlying user intention. Following this, attribute extraction, conditioned on the task classification, extracts attribute values that are necessary for the expert tools to address the causal problem. Instead of formulating it as a sequential process, we propose to merge them into one sequence-to-sequence (seq2seq) procedure where the output is a structured JSON with the "causal_problem" key and other task-specific keys, e.g. "dataset" ($\mathcal{D}$), "treatment" ($\mathcal{A}$), "outcome" ($\mathcal{Y}$). Some simulated input queries with the corresponding JSON output are shown in Appendix C.2 Table 6.

Even with careful prompt engineering, it is challenging to directly adapt a pre-trained LLM for such a seq2seq task, where the latest GPT4 checkpoint still has a 31% error rate (see Section 4.3 for details) and publically available model such as Llama 2 failed on this JSON extraction task. As a result, we find it necessary to fine-tune a local LLM with an augmented dataset consisting of both input queries and output JSONs.

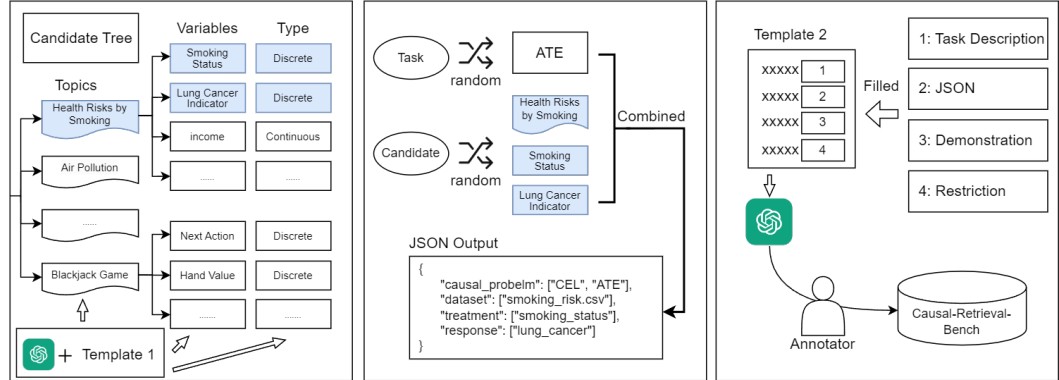

Figure 3: Causal-Retrieval-Bench construction procedures for the first step. GPT prompts used in this section can be found in Appendix C.2. In the left panel, GPT is prompted to list out different topics and measurable variables for each topic. With randomly drawn tasks from section 2 and some variables under the same topic, we generated JSON output in the middle panel. The colored boxes in the left and middle panels are shared topics and variable names of interests. In the right panel, the numbered boxes in template 2 mean blank places to be filled with the task description, the supplied demonstration, JSON inputs, and output restrictions. Common connectors and general instruction in prompt 2 are shortened into "xxxxx" due to the figure limit.

### 3.1.1 Construct Causal-Retrieval-Bench

To the best of our knowledge, the aforementioned customized input-output pairs can not be collected from the online corpus, since the commonly used LLM training data are mainly conversational materials other than structured data. Therefore, we choose to construct a customized corpus containing both synthetic causal questions and the corresponding JSON representations by prompting an instruction-following LLM.

To collect LLM-generated data, the common approach starts by writing several demonstrated examples. Those demonstrations are then fed to an LLM (e.g., GPT-4) for new sample generation. We have observed that such an approach suffers from (1) non-controllable topics and variables of interest, which results in content homogeneity, (2) incomplete/non-compliant JSON outputs, and (3) inappropriate paraphrase, where the causal question is transformed to be association-related. To resolve the pain points and prevent data contamination, we propose a three-course procedure, which is illustrated in Figure 3.

The procedure can be summarized as an output-first strategy, where the major goal is to improve the correctness and the variety of the output JSON. An instruction prompt (See Appendix C.2) is then incorporated to construct causal-related questions in different sentence structures. Specifically, we first prepare a topic database by conducting value generation, then generate JSON data by random sampling, and finally paraphrase the question utilizing prompting.

- **Value Generation**: the goal of the first course is to get a candidate pool of the interested topic and variables. We prompt the GPT model to generate some potential topics and related variable names. For each variable, we perform a zero-shot classification by GPT to determine the variable type, i.e., whether it is discrete or continuous. All the collected information is maintained in a three-level hierarchy structure (topic -> variable -> type).

- **JSON Generation**: Based on the assumption that causality tasks often involve variables within the same topic, we randomly generate structured JSON data following the topic-variable-type structure. For example, for an average treatment effect estimation task, a topic name is firstly drawn from the stem of the tree as the data file name, and two variables underneath are then considered as candidates for the synthetic treatment and outcome. For causal tasks where a specific value/condition

of a group variable is needed, we sampled a random value, either continuous or discrete, based on the corresponding variable type.

- **Question Generation**: For each JSON output, we prompt the GPT-4 model to generate five related questions given the context. Utilizing a univariate prompt template, we transform each JSON into an information-augmented prompt consisting of task-specific descriptions, expert-written demo examples, JSON information, and paraphrasing guidelines. More details can be found in Appendix C.2

During development, the aforementioned three courses are executed sequentially with quality control on the correctness of generated entities/queries in between. Confused with other statistical concepts, such as correlation and association, the current GPT model has suboptimal performance on the causality-related task. Hence, domain experts are involved after each step to improve the data quality with human annotation. As a result, for the Causal-Retrieval-Bench, we collect 1,500 pairs, i.e., 300 for each causal task, of causal questions with the corresponding JSON outputs. The detailed approach for model fine-tuning is move to D.1 for brevity.

## 3.2 Step2. assign and execute causal tools

In this stage, to address user queries, we first select suitable causal learning algorithms based on the task class determined in Step 1. Following this selection, the algorithm is executed automatically, using both the extracted information from Step 1 and the user-provided dataset as inputs, to get the estimated result. See Figure 4 in Appendix C.3 for a graphical illustration. Depending on the causal task, the output result can be a causal graph $\mathcal{G}$, numerical values of estimated causal effect, or a treatment level $a \in \mathcal{A}$. LLM4Causal supports a variety of approaches for each causal task by leveraging well-known public causal packages, including econml(Oprescu et al.), causalml(Chen et al., 2020), causal-learn(Zheng et al., 2023), and CausalDM(Song et al., 2023). Additionally, we remark that LLM4Causal is designed with flexibility, allowing for the incorporation of new methodologies through simple integration of function scripts.

## 3.3 Step3. interpret the numerical output

As discussed in Section 3.2, Step 2 produces a numerical value of causal effect, a recommended action level, or an adjacency matrix indicating causal relations in a causal graph. Interpreting these outputs needs domain expertise and may not be easily understood by a general audience lacking a causal background. To enhance user-friendliness and reduce the learning curve of LLM4Causal, our final stage entails the interpretation of the function output using fluent natural language. The process starts with a templated interpretation that automatically converts the numerical output to a one-sentence summary (See Appendix C.4.1 for examples). The final human-readable interpretation is then produced through the utilization of the LLM4Causal, which is guided by a designed prompt that includes response guidelines, the templated interpretation, and other relevant information (i.e., selected tools, the original user query, etc.). The prompt instructs the LLM4Causal to avoid generating hallucinatory content, ensure the inclusion of all relevant information, and provide cohesive responses in the relevant context of the original question (See Appendix C.4.3 for details). Observing that even GPT4 produces problematic interpretations (25% hallucination; see the definition below), we further fine-tune the LLM4Causal for output interpretation. In the following, we outline the process used to generate the golden dataset Causal-Interpret-Bench used for model calibration.

### 3.3.1 Build Causal-Interpret-Bench

Likewise to the procedures in Step 1, we first use the GPT4 model to generate a silver dataset of interpretations, employing the templates in Appendix C.4.1 and the prompts in Appendix C.4.3. The interpretation instruction (prompt) takes the original causal question and intermediate results from steps 1 and 2 as inputs. Among them, user queries and their corresponding causal task classifications from section 3.1.1 are directly reused, with

causal tools paired using the methodology described in 3.2. As for the function outputs, the numerical values are randomly generated in formats specific to each task. To create the final golden dataset, we then have annotators manually revise 400 interpretations, avoiding the three types of mistakes defined in the evaluation rubric outlined below:

- **Hallucination**: i) The interpretation incorrectly presents information, such as treating the response variable as the treatment variable and misinterpreting the direct effect in mediation analysis as the total effect; ii) it includes details that are not present in the provided context, such as commenting on the effectiveness of methods used; or iii) it incorrectly applies terms such as 'correlation' and 'association', which are inappropriate for describing causal relations.

- **Incompleteness**: The interpretation lacks one or more essential components, such as the data set, the method, the results, and / or all the variables involved.

- **Non-fluency**: i) The interpretation includes unexplained, meaningless variable names (e.g., 'chg_rate'); ii) it repeats the same content multiple times; or iii) it directly references or rephrases the guidelines in the prompt.

Although various metrics are developed to assess the factual consistency between output summaries/interpretations and the original context (Zhang et al., 2019), as well as the conciseness and readability of output summaries, such as the ROUGE score (Lin, 2004), they do not meet our specific needs. The interpretation we require is highly specialized, necessitating a clear distinction between causality and association, guaranteeing that no crucial information is missed, and ensuring it is comprehensible to general audiences, among other criteria. Furthermore, the current automatic evaluation based on sentence similarity metrics or GPT-based prompts fails to adequately incorporate human preferences, making them unsuitable for our application. Therefore, instead, we carefully defined the three aforementioned rubrics to guide human annotation.

## 4 Experiments

In this section, we conduct numerical experiments to investigate the performance of LLM4Causal on five causal decision-making tasks. For each causal task defined in Section 2, we randomly sample topics and their related variables following the process described in Figure 3 that is different from the two aforementioned datasets. With the topics and variables in hand, we further generate 150 synthetic data files utilizing the methodology introduced in Section D.2. Based on these synthetic datasets, we evaluate end-to-end performance for the proposed LLM4Causal model in Section 4.2. Furthermore, by comparing with the partially-capable benchmark methods, i.e. GPT4, in each step, detailed ablation analyses are conducted to provide a comprehensive understanding of the model performance in Section 4.3 and 4.4.

### 4.1 Benchmark models

The Large language models for evaluation are described as follows.

- **GPT4** (OpenAI, 2023b): For end-to-end evaluation, we tested ChatGPT with a GPT4 backbone. GPT4 with the function call feature enabled are tested for ablation analysis. By creating a function with causal task description and input arguments requirements, we utilize the GPT 4 with the function calling feature to generate the output, which is then transformed into JSON format based on the predicted "causal_problem" key.

- **LLM4Causal**: A customized language model started from Llama-2 (7B) checkpoint (Touvron et al., 2023b) and further fine-tuned on the Causal-Retrieval-Bench and Causal-Interpret-Bench. To inspect if mixing two types of data will harm the performance of each other, we present three varieties where LLM4Causal-Mixed trained on both datasets, LLM4Causal-Retrieve trained on the retrieval bench, and LLM4Causal-Interpret augmented by the interpret bench.

We only have the GPT family as the benchmark as other closed-sourced LLM environments do not have code execution ability. Furthermore, existing works (Qin et al., 2023b; Yang et al., 2023) on tool calling do not support causal tasks or fully rely on prompting pre-trained models for output interpretations. Preliminary experiments (see Appendix D.3) show that pre-trained open-sourced models, such as Llamma2-7B, fail to adequately handle prompts with our designed requirements, making the aforementioned function-calling methods incapable of finishing the designed tasks.

## 4.2 Main result: end-to-end evaluation

In this experiment, we evaluate the end-to-end model performance from receiving causal-related questions to providing interpretations that explain the causal results. With such a setup, there is no sole "golden label" existing for each question as the correct interpretation is not unique. To provide a comprehensive analysis, we provide the following evaluation metrics inspired by previous works (Yang et al., 2023; Qin et al., 2023b) from three aspects:

- **Pass rate** (Qin et al., 2023b) calculates the proportion of user requests that could be completed by the model. This metric measures the model's executability and could be calculated as Pass rate $= \frac{1}{N} \sum_{i=1}^{N} \mathbb{I}\{\tau_i\}$, where $N$ is the number of samples, $\mathbb{I}\{\tau_i\}$ is an indicator function, which equals to 1 if the model could generate an output for the $i^{th}$ user request, and 0 otherwise.

- **Relevance rate**: it calculates the proportion of user requests that the model provides relevant content with the correct causal task. This metric measures the model's task identification ability as Relevance rate $= \frac{1}{N} \sum_{i=1}^{N} \mathbb{I}\{\gamma_i\}$, where $\mathbb{I}\{\gamma_i\}$ is an indicator function, which equals to 1 if the model could generate an output that correctly identify the causal task for the $i^{th}$ user request, and 0 otherwise.

- **Win rate**: it calculates the proportion of user requests that the model has delivered an accurate result. It could be calculated as Win rate $= \frac{1}{N} \sum_{i=1}^{N} \mathbb{I}\{\eta_i\}$, where $\mathbb{I}\{\eta_i\}$ is an indicator function which equals to 1 if the model output contains the desired true value for the $i^{th}$ user request, and 0 otherwise.

We evaluated the end-to-end causal-question answering performance for ChatGPT and LLM4Causal-Mixed in Table 2. The configuration of the ChatGPT can be found in Appendix D.4.

|  | ChatGPT | | | | | LLM4Causal-Mixed | | | | |
|---|---|---|---|---|---|---|---|---|---|---|
|  | CGL | ATE | HTE | MA | OPO | CGL | ATE | HTE | MA | OPO |
| Pass Rate | 0.17 | 0.77 | 0.73 | 0.27 | **0.87** | **1.00** | **0.93** | **0.83** | **0.86** | 0.83 |
| Relevance Rate | 0.10 | 0.60 | 0.43 | 0.20 | 0.43 | **1.00** | **0.93** | **0.83** | **0.80** | **0.83** |
| Win Rate | 0.00 | 0.37 | 0.07 | 0.10 | 0.07 | **0.90** | **0.90** | **0.80** | **0.70** | **0.73** |

Table 2: End-to-end evaluation for LLM4Causal-Mixed. *Higher is better.*

LLM4Causal provides a high pass rate and relevance rate for all tasks, whereas GPT4 performs well regarding pass rate but shows dichotomized performance in Relevance Rate. The pass rate only requires the model to successfully execute its last code block without an error, regardless of relevance. It is important to highlight that passing cases for GPT4 include cases where GPT only printed the basic statistics of the data without in-depth analysis. For tasks having a high Pass Rate, such as HTE and OPO, we got 42% and 51% of them to be either superficial starter conversations or irrelevant outputs that confuse general audiences.

More importantly, LLM4Causal outperforms ChatGPT on Win Rate. On the one hand, we can find that ChatGPT with the GPT-4 backbone can only finish 37% of the ATE task and nearly failed all other scenarios. The main reason is the generated code either has syntax/factual errors or attempts to load a package that is not in the environment. On the other hand, LLM4Causal correctly answered 80.6% questions on average. The win rate roughly reflects the combined difficulty of three steps for each task. The MA and OPO tasks

require more parameters inferred from the question, further boosting the task complexity and resulting in a relatively lower win rate.

By investigating the failed cases, we observe that some GPT4 responses to CGL tasks are still informative to users but fail due to a lack of code dependencies. To further analyze the root cause, we conducted two more ablation analyses on steps 1 and 3 to strip out the cases when code execution is the main roadblock. Meanwhile, we compare the LLM4Causal-Mixed to LLM4Causal-Retrieve/LLM4Causal-Interpret in the ablation analysis to explore whether training distinct models on each dataset could enhance performance.

### 4.3 Ablation analysis 1: causal entity extraction

In this section, we focus the model performance on Step 1, user request interpretation, comparing with the benchmark methods on the synthetic data of 150 causal-related questions. In detail, for each simulated user request, the model is asked to produce a JSON output following the output format listed in Table 6. The accuracy of each key value is then calculated by comparing the ground true label with the model outputs. It is worth mentioning that, we require an exact match for values of the causal task key and the dataset key since the causal task lists are provided in the training data and the dataset names are explicitly mentioned in the user request. For the remaining key values, we consider the value to be correctly retrieved if it is a subset of the model output (soft-match), e.g. extracting both "customer_satisfaction_rate" or "satisfaction" would be considered as correct if the true label is "satisfaction_rate". We used the GPT4 with the function call feature enabled as a benchmark method, as most existing studies (Qin et al., 2023b; Zhang, 2023) treat its output as the golden label for local LLM development, and the detailed setup can be found in Appendix D.4.

| Metric | Causal Task | Dataset | Nodes | Treatment |
|---|---|---|---|---|
| GPT4-turbo | 0.69(0.02) | **0.99**(0.01) | **0.99**(0.02) | 0.60(0.02) |
| LLM4Causal-Retrieve | 0.96(0.00) | **0.99**(0.01) | 0.88(0.03) | 0.94(0.00) |
| LLM4Causal-Mixed | **0.98**(0.00) | **1.00**(0.00) | **1.00**(0.00) | **0.96**(0.00) |
| Metric | Response | Mediator | Condition | **All** |
| GPT4-turbo | 0.60(0.02) | 0.61(0.04) | 0.98(0.01) | 0.77(0.01) |
| LLM4Causal-Retrieve | 0.94(0.01) | 0.90(0.05) | **1.00**(0.01) | 0.96(0.01) |
| LLM4Causal-Mixed | **0.97**(0.00) | **1.00**(0.00) | **1.00**(0.00) | **0.98**(0.00) |

Table 3: Causal entity extraction performance. *Higher is better.*

As shown in the table 3, both LLM4Causal models can effectively solve step 1 and significantly outperform the GPT4-turbo. This is dominated by the accuracy of the causal task identification, which blocks the model from accurately finding the necessary entities to extract if a wrong task class is posited. Besides, a seemingly surprising fact is that the mixed version even outperforms the retrieve-only version on all tasks. Such a phenomenon may be due to the inclusion of the interpretation data, which also involves causal queries and their corresponding causal tasks with interpretations, increasing the parameter weight during the fine-tuning process of the model between task-specific queries and the corresponding task classification.

### 4.4 Ablation analysis 2: interpreting causal function output

Following metrics described in section 3.3.1, we evaluate the quality of model-generated interpretations. A double-blind experiment is conducted with domain experts judging if the interpretation generated by each language model has issues with hallucination, fluency, or incompleteness using the same template. The averaged error rate is reported in Table 4.

The interpretation results showed that LLM4Causal Models are comparable to or superior to the GPT-4-turbo on both hallucination and fluency metrics. Compared with the initial pre-trained Llama2, the model fine-tuning procedure successfully instructs it to reduce

| model | task | hall. | inco. | fluent | task | hall. | inco. | fluent |
|---|---|---|---|---|---|---|---|---|
| GPT4-turbo | | 0.08 | **0.00** | 0.28 | | **0.05** | **0.10** | 0.22 |
| LLM4Causal-Interpret | CGL | **0.05** | 0.03 | 0.28 | MA | 0.10 | 0.15 | 0.18 |
| LLM4Causal-Mixed | | 0.23 | **0.00** | **0.20** | | 0.19 | 0.27 | **0.16** |
| GPT4-turbo | | 0.43 | 0.05 | 0.48 | | **0.15** | **0.10** | 0.33 |
| LLM4Causal-Interpret | ATE | 0.38 | 0.10 | **0.20** | OPO | 0.26 | 0.13 | 0.18 |
| LLM4Causal-Mixed | | **0.28** | **0.00** | 0.50 | | 0.21 | 0.13 | **0.10** |
| GPT4-turbo | | 0.08 | **0.00** | 0.28 | | 0.19 | **0.06** | 0.36 |
| LLM4Causal-Interpret | HTE | **0.05** | 0.03 | 0.28 | **All** | **0.17** | 0.10 | **0.20** |
| LLM4Causal-Mixed | | 0.23 | **0.00** | **0.20** | | 0.20 | 0.08 | 0.28 |

Table 4: Causal result interpretation performance, where hall. indicates hallucination, inco. stands for incompleteness, and fluent is for non-fluency. *Lower is better.*

hallucination and makes the interpretation crisp without including sentences repeating already-stated information. Besides, we expect our fine-tuned models to be improved, especially for the completeness metric, by involving more golden samples for fine-tuning.

While the direct result shows the interpret-only model outperforms the mixed version, the improvement in hallucination and fluency from gpt4 to LLM4Causal-Mixed still shows the promise of calibrating a single LLM4Causal model as a more economical solution.

## 5 Conclusion

In this paper, we have proposed LLM4Causal, the first end-to-end user-friendly large language model with causal decision-making ability. Without requiring in-depth domain knowledge, the proposed model could be easily used for the general audience, which remedies the weakness of the current LLM application on the causal tasks. Furthermore, we have proposed a novel data generation pipeline by utilizing GPT-based LLM models and human annotations to improve the data quality in terms of both data variety and accuracy. From the numerical experiments, we have shown that calibrating a single LLM to accomplish such an end-to-end task is possible. Fine-tuned to interpret users' query and function outputs, LLM4Causal has shown superior performance in causal entity extraction and function result interpretation.

To conclude, we briefly discuss some limitations followed by possible future directions. First, the proposed model focuses on five major causal tasks, and it is worth mentioning that our framework could be easily extended to support more causal tasks with more methodologies. Second, in this paper, we only leverage LLM's tool-usage capabilities to utilize statistical tools for data analysis. Concurrently, the internal knowledge acquired by the LLM during its pre-training process has proved advantageous for common sense retrieval. The integration of the LLM's tool-usage capabilities with its inherent knowledge base could enhance its performance on statistical causal tasks. Furthermore, integrating LLM's internal knowledge for general reasoning might enable the model to discern textual causality between descriptive events in the conversation (Rashkin et al., 2018; Ning et al., 2019; Wang et al., 2023), a concept related to, yet distinct from, the causality examined in this paper. Lastly, a potential direction to explore is the interactive potential of the framework so that the proposed method could iteratively adapt to users' feedback, which could be beneficial in following users' preferences for causal tools, encouraging users to collect necessary information when their input is not enough, and enhancing the interpretations based on their extra input in the problem context.

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

# A  Related work

In this section, we briefly review the literature that are closely related to our paper, which includes causal decision-making methods, LLM with causal capabilities, enhancement methods for pre-trained LLM, as well as large language models with tool usage.

## A.1  Causal decision making

The decision-making process, crucial in diverse real-life contexts, fundamentally relies on understanding counterfactual scenarios, thereby emphasizing the significance of causal analysis in decision-making. Recently, Song et al. (2023) proposed a comprehensive framework known as Causal Decision Making (CausalDM, Song et al., 2023), which examines various decision-making stages through a causal lens and establishes connections among the relevant literature under the same conceptual umbrella. In general, the body of current literature can be classified into three basic categories: i) causal structure learning (CSL) (Spirtes et al., 2000; Shimizu et al., 2006; Zheng et al., 2018; Yu et al., 2019; Glymour et al., 2019), which focuses on unraveling the complex causal relationships among variables, is pivotal for pinpointing vital factors that influence outcomes, either directly or indirectly; ii) causal effect learning (CEL) (Yao et al., 2021; Hicks & Tingley, 2011), which is primarily concerned with quantifying the effects resulting from different causal paths within a known causal structure, typically concentrating on measuring the difference between outcomes of receiving versus not receiving treatments; and iii) causal policy learning (CPL), which involves evaluating various treatment strategies and developing optimal action plans, frequently investigated as dynamic treatment regime problems (Chakraborty & Murphy, 2014) or conceptualized within the framework of reinforcement learning (Sutton & Barto, 2018; Zeng et al., 2023).

## A.2  Large language models with causal capabilities

The rapid development of LLMs has inspired some innovative methods for causal graph discovery. Kıcıman et al. (2023) is one of the pioneering works in this field, introducing a method that relies solely on "meta-data" (variable names) instead of specific observed variable values. Using LLMs like GPT-3.5 and GPT-4, this approach detects causal relationships between variable pairs through direct querying. Similarly, Long et al. (2023) use GPT-3 to discover causal relationships from variable names within the medical domain, motivated by the capacity of LLM to extract knowledge from the medical texts on which it was trained. However, as Long et al. (2023) highlighted, the efficacy of this type of approach relies heavily on the prompts used for LLM queries and the initial training corpus of the LLM. In a further development, Ban et al. (2023) proposes to integrate the causal relations retrieved directly from LLMs and traditional score-based causal discovery methods. In their framework, the LLM's causal relations are utilized to guide the optimization process of score-based methods as soft or hard constraints, enhancing the performance compared to using either LLMs or traditional methods alone. Gupta et al. (2023), on the other hand, apply LLMs in selecting causal discovery methods for a specific dataset rather than directly discovering the causal relations on the dataset given. Their framework is claimed to be able to select the causal discovery method with the best F1 score out of a fixed set of candidates. Furthermore, Zhang et al. (2023) discusses the strengths and limitations of LLMs for causal discovery. Although LLMs have advantages in detecting causal relationships within the scope of their training data, their performance suffers when confronted with unfamiliar domains. It is worth mentioning that our approach differs from the previously mentioned studies, as none of them introduces a mechanism to automatically invoke data-based causal discovery methods, nor do they have a mechanism to translate the output into natural languages that general audiences can understand.

## A.3  Enhancing pre-trained large language models

Pre-trained large language models often contain billions of parameters that impose challenges when fine-tuning the entire model to adapt to downstream tasks. Two classes of

research are developed to reduce such a computational burden. A straightforward but effective workaround is freezing most parts of the model and only tuning a small portion of the parameters, which includes additive methods and reparametrization-based methods (Lialin et al., 2023). Additive methods, such as adapters (Houlsby et al., 2019), augment the pre-trained model with extra parameters and only train the newly added parameters. Reparametrization-based methods, such as LOw-Rank Adaptation method (LoRA) (Hu et al., 2021), reduce the number of trainable parameters utilizing low-rank representations. To improve the training efficiency, our proposed method utilizes the LoRA method (Hu et al., 2021), which reduces trainable parameters by freezing the pre-trained model and optimizing the rank-decomposition matrices.

Another line of research has demonstrated that template-based prompting can also notably enhance the performance of LLMs. Commonly used prompt types include cloze question style prompts, which consist of string templates with placeholders for the LLM to fill in (Rajagopal et al., 2021; Cui et al., 2021), and prefix style prompts that prepend tokens to the input texts of LLMs (Li & Liang, 2021; Lester et al., 2021). The latter is particularly effective for generative tasks, such as generating interpretations of numeric outputs from causal models in our LLM4Causal framework (Liu et al., 2023). A common way to create such prompts is to design them manually and such prompts usually include a few human-generated exemplars of input-output pairs for LLMs (Chung et al., 2022). These human-crafted prompts have improved performance across various tasks such as question answering and text classification (Brown et al., 2020; Schick & Schütze, 2020). Recently, Wei et al. (2022) demonstrated that prompting language models to add reasoning steps before the actual output significantly enhances the LLM's performance on arithmetic, common-sense, and symbolic reasoning tasks. To reduce manual effort in this process, some studies have also explored automation in prompt design. For instance, Shin et al. (2020) discretely optimized the prompt by adding tokens from a certain word collection, while Haviv et al. (2021) focuses on automatic paraphrasing of given manually designed prompts and selecting the most effective one from these variations. In our LLM4Causal framework, we utilize prefix style prompts with manually designed input-output examples to effectively collect data for fine-tuning LLMs to classify user queries into different causal tasks and interpret the numerical results from causal tools. The satisfying performance of our framework has demonstrated the effectiveness of our carefully crafted prompts.

### A.4 Large language model with tool usage

Despite LLM's impressive performance in natural language processing tasks, some inherent limitations remain. Examples include the inability to perform precise arithmetic operations (Patel et al., 2021), lack of access to up-to-date information (Komeili et al., 2021), and ignorance of temporal contexts (Dhingra et al., 2022). To overcome these impediments, a natural solution is to enable LLM to utilize tools, such as calculator, search engine, and calendar, to accomplish complex tasks (Qin et al., 2023a). In particular, Schick et al. (2023) proposed Toolformer, a self-supervised language model that could use external tools by calling relevant APIs, without relying on large human-annotated data. With a few tool-usage demonstrations in the instruction prompt, Toolformer is trained to learn which, when, and how APIs could be called to complete a variety of tasks, including question answering, Wikipedia search, and mathematical reasoning. Based on the toolformer architecture, Graph-Toolformer Zhang (2023) is proposed to teach LLMs to use tools on graph reasoning tasks with ChatGPT-augmented prompts (OpenAI, 2022). Moreover, GPT4tools (Yang et al., 2023) are proposed to facilitate LLMs in accomplishing visual-related tasks by generating prompts with multi-modal contexts, and ToolLLM (Qin et al., 2023b) further generalizes the tool-use capability of LLMs to master over 16,000 real-world APIs and enable multiple tool-usage for each query. Focusing on developing frameworks for calling general Application Programming Interfaces (APIs), these methods cannot be directly applied to causal tasks without a well-developed causal API, which currently is not available. On the contrary, our proposed approach is specifically designed for causal tasks based on a framework that is most closely related to Toolformer.

# B    Detailed problem formulation

In this paper, we aim to augment existing pre-trained LLMs with the proficiency to address causal inquiries, thereby opening the door to causal decision-making processes for general audiences. As introduced previously, causal decision-making tasks can be classified into three categories: 1) causal structure learning (CSL), 2) causal effect learning (CEL), and 3) causal policy learning (CPL). Each of these primary categories consists of multiple tasks, and our study focuses on five key tasks that are particularly relevant to the interests of our target audience. Let us denote the user query as $\mathcal{Q}$, the dataset to be analyzed as $\mathcal{D}$, the node set whose interrelationship is of interest as $\mathcal{X}$, the treatment variable as $\mathcal{A}$, the response variable as $\mathcal{Y}$, the mediator variables as $\mathcal{M}$, and the condition of the subgroup of interest as $\mathcal{S}$. The five causal tasks are listed as follows and summarized in Table 5.

| Task | Dataset $(\mathcal{D})$ | Nodes $(\mathcal{X})$ | Treatment $(\mathcal{A})$ | Response $(\mathcal{Y})$ | Mediator $(\mathcal{M})$ | Condition $(\mathcal{S})$ | Function Output Format |
|------|---------|-------|-----------|----------|----------|-----------|------------------------|
| CGL | ✓ | ✓ | | | | | Causal Graph $(\mathcal{G})$ |
| ATE | ✓ | | ✓ | ✓ | | | Numeric Value |
| HTE | ✓ | | ✓ | ✓ | | ✓ | Numeric Value |
| MA | ✓ | | ✓ | ✓ | ✓ | | Numeric Values |
| OPO | ✓ | | ✓ | ✓ | | ✓ | Treatment Level |

Table 5: Constructions of causal tasks

**[CSL] Causal Graph Learning (CGL):** Causal graph learning, as its name suggests, involves discovering the causal relationships among variables. This task is crucial for addressing causal queries such as identifying causal links between pairs of variables, determining the variables that influence a particular variable of interest, and quantifying the number of causally connected variable pairs. Given $\mathcal{Q}$ and $\mathcal{D}$, our goal is to learn and report a directed graph, denoted as $\mathcal{G}$, that encapsulates the entire causal structure among variables in $\mathcal{X}$, which is either explicitly specified in $\mathcal{Q}$ or, by default, encompass all variables in $\mathcal{D}$.

**[CEL] Average Treatment Effect Estimation (ATE):** When the treatment and the target response variable have already been determined, the primary interest lies in quantifying the intervention's effect size. The objective of the ATE task is to evaluate the average effect of $\mathcal{A}$ on $\mathcal{Y}$ by measuring the difference in counterfactual outcomes between the treated group and the control group. Let $\mathcal{Y}(1)$ denote the response that would be observed if the treatment was applied, and $\mathcal{Y}(0)$ denote the response that would be observed if no treatment is applied. Using the provided dataset $\mathcal{D}$, the treatment variable $\mathcal{A}$, and response variable $\mathcal{Y}$ defined in $\mathcal{Q}$, our objective is to execute appropriate ATE learners to calculate the $\mathbb{E}[\mathcal{Y}(1) - \mathcal{Y}(0)]$.

**[CEL] Heterogeneous Treatment Effect Estimation (HTE):** Similar to the ATE tasks, the HTE tasks also aim to measure effect sizes. However, HTE tasks diverge by focusing on generating nuanced insights tailored to specific subgroups or individuals, characterized by specific conditions. Using the dataset $\mathcal{D}$, and given the treatment variable $\mathcal{A}$, response variable $\mathcal{Y}$, and specific conditions $\mathcal{S}$ as outlined in $\mathcal{Q}$, we aim to call suitable HTE learners to evaluate the conditionally expected difference $\mathbb{E}[\mathcal{Y}(1) - \mathcal{Y}(0)|\mathcal{S}]$.

**[CEL] Mediation Effect Analysis (MA):** Mediators, which act as channels for transmitting treatment effects to the response, are prevalent in various real-world applications. With the provided dataset, and given the $\mathcal{A}$, $\mathcal{Y}$, and $\mathcal{M}$ specified in $\mathcal{Q}$, MA's objective is to calculate both the direct effect $\mathbb{E}[\mathcal{Y}(1, \mathcal{M}(0)) - \mathcal{Y}(0)]$ and the indirect effect $\mathbb{E}[\mathcal{Y}(1) - \mathcal{Y}(1, \mathcal{M}(0))]$. Here, $\mathcal{M}(\cdot)$ is analogously defined to $\mathcal{Y}(\cdot)$, and $\mathcal{Y}(a, \mathcal{M}(a'))$ denote the response that would be observed if treatment $a$ was applied and the mediator set to the level that would be observed if treatment $a'$ was applied.

**[CPL] Off-Policy Optimization (OPO):** As the name suggests, OPO task is a one-stop shop for decision-makers. The goal of OPO is to identify the optimal action (i.e., treatment level) that optimizes the expected response to be obtained given the current circumstances. Utilizing the dataset $\mathcal{D}$ provided, with $\mathcal{A}$, $\mathcal{Y}$, and $\mathcal{S}$ explicitly specified in $\mathcal{Q}$, we aim to select suitable policy learners to determine the optimal action $a^* = argmax_a \mathbb{E}(\mathcal{Y}(a)|\mathcal{S})$, which

would then be recommended to the user. We can provide either single-stage or multi-stage recommendations depending on the data structure of $\mathcal{D}$.

To tackle the aforementioned diverse tasks using a single LLM, we introduce LLM4Causal. By fine-tuning pre-trained LLMs, LLM4Causal is capable of comprehending causal queries, applying appropriate causal tools to analyze the provided tabular dataset, and providing answers by interpreting numerical results in straightforward and fluent language. More technical details of LLM4Causal is discussed in Section 3.

## C  Supplimentary methodology

### C.1  Demonstration input-output pairs

Following procedures described in 3.1, which condition on the task classification, extract attribute values that are necessary for the expert tools to address the causal problem. For each input question, the corresponding output is a structured JSON with the "causal_problem" key and other task-specific keys, e.g. "dataset" ($\mathcal{D}$), "treatment" ($\mathcal{A}$), "outcome" ($\mathcal{Y}$). Some simulated input queries with the corresponding JSON output are shown in Table 6.

| Task | Example Input Query | Expected JSON Output |
|---|---|---|
| CGL | Does the disaster_risk_reduction.csv dataset provide evidence of a direct link between building code compliance rate and the effectiveness of disaster preparedness campaigns? | {'causal_problem': ['CSL', 'CGL'], 'dataset': ['disaster_risk_reduction.csv'], 'nodes': ['building_code_compliance_rate', 'disaster_preparedness_campaigns']} |
| ATE | How does the labor force participation rate (labor_participation_rate) in employment.csv contribute to changes in wage growth (wage_increase)? | {'causal_problem': ['CEL', 'ATE'], 'dataset': ['employment.csv'], 'treatment': ['labor_participation_rate'], 'response': ['wage_increase']} |
| HTE | Based on the findings in the cybersecurity.csv dataset, what impact does the presence of data breach incidents have on cybersecurity investment under a group condition where the cybersecurity readiness index is set at 0.5 (readiness_index=0.5)? | {'causal_problem': ['CEL', 'HTE'], 'dataset': ['cybersecurity.csv'], 'treatment': ['data_breach_incidents'], 'response': ['cybersecurity_investment'], 'condition': [('readiness_index', 0.5)]} |
| MA | Is there substantial evidence in retail.csv indicating that the pathway from retail employment to the e-commerce penetration rate is mediated by the consumer confidence index? | {'causal_problem': ['CEL', 'MA'], 'dataset': ['retail.csv'], 'treatment': ['retail_employment'], 'response': ['e-commerce_penetration_rate'], 'mediator': ['consumer_confidence_index']} |
| OPO | If the poverty rate stands at 0.32 (poverty_ratio = 0.32), what recommendations can be derived from the poverty.csv dataset on adjusting social assistance coverage to positively impact the gini coefficient? | {'causal_problem': ['CPL', 'OPO'], 'dataset': ['poverty.csv'], 'treatment': ['social_assistance_coverage'], 'response': ['gini_coefficient'], 'condition': [('poverty_ratio', '0.32')]} |

Table 6: Example Query Inputs with expected JSON Outputs

### C.2  Details about step 1

The overall cloze template contains 5 different missing pieces. Among them, the task requirement is a verbal description of the task like the role definition of ChatGPT. JSON information is followed by listing out each item line by line. Further, some randomly drawn human-written samples, split into two parts, are added as demonstrations for the GPT

model to further understand what kind of output we need. Finally, we add the requirements for the output in the last part for better control of the GPT output.

```
Given the details below, generate five diverse and plain-worded questions
[Task requirements]:
- [JSON information]
-----------------------------------------------------------
Here are some examples:
Example 1: given the following information:
- [Demonstration 1 part A]

The corresponding questions can be:
- [Demonstration 1 part B]

-----------------------------------------------------------
[Requirements]
```

Below is an example of the full instruction for Causal Structure Learning data generation with information filled in:

```
Given the details below, generate five diverse and plain-worded questions
asking about the existence
of causal effects between interested variables:
- Dataset name: social_media.csv
- Interested variable: number of social media users, user demographics
-----------------------------------------------------------
Here are some examples:
Example 1: given the following information:
- Dataset name: employee_data.csv
- Interested variable: all variables

The corresponding questions can be:
- Is there a method to discover every direct influence present in
the employee_data.csv dataset?
- Within the employee_data.csv dataset, how many instances of one factor
directly causing another
can be observed?
- How many causal connections can be identified in the
employee_data.csv dataset?

Example 2: given the following information:
- Dataset name: job_data.csv
- Interested variable: education (edu), job satisfaction
(job_satisfaction)

The corresponding questions can be:
- Within the job_data.csv dataset, is there causal links between education
(edu) and job satisfaction
(job_satisfaction)?
- Does the job_data.csv data reveal any direct relationships between
education (edu) and job
satisfaction (job_satisfaction)?
- Are there discernible causal links between education (edu) and job
satisfaction (job_satisfaction) in the job_data.csv dataset?

Example 3: given the following information:
- Dataset name: weather_data.csv
- Interested variable: temperature (temp), humidity (humid), rainfall
(rain), local latitude (lat), local altitude (alt)
```

```
The corresponding questions can be:
- What are causal links among temperature (temp), humidity (humid),
rainfall (rain), local latitude (lat), and local altitude (alt) in dataset
weather_data.csv?
- Within the dataset weather_data.csv, what is the nature of the causal
links among temperature (temp), humidity (humid), rainfall (rain),
local latitude (lat), and local altitude (alt)?
- In the weather_data.csv dataset, what causal connections can be
identified among temperature (temp), humidity (humid), rainfall (rain),
local latitude (lat), and local altitude (alt)?

---------------------------------------------------------
Ensure that the questions employ a mix of different phrasing and diverse
sentence structures.
Ensure that the questions do not mention correlation and association.
Ensure that the questions place a strong emphasis on the effect existence
or the total number.
Ensure that all variable names under interested variables field exist in
all five quetsion.
Ensure that the provided interested variable names are integrated
naturally into the questions without discarding or altering any part of them.
```

Specifically, we implement GPT prompts in this step by a modulized template to improve components' reusability. Structurally speaking, the prompt starts with a description of the question generation task and the corresponding causal task. It is then followed by information extracted from a JSON from the previous course as the contextual component for the generated question. Subsequently, randomly sampled demonstrations are included as the sample illustrations and output restrictions are itemized as the final remark.

Using the example of CGL task in Table 6 as an illustration, the output JSON has two task-specific keys conditioned on its inferred classification as a CGL task: "dataset" with the value "disaster_risk_reduction.csv," and "nodes", which includes 'building_code_compliance_rate' and 'disaster_preparedness_campaigns'.

### C.3 Details about step 2

Figure 4 describes the procedure of step 2 workflow, where we utilize the output from step 1 to select an appropriate input function as well as its necessary input fields to conduct causal analysis.

### C.4 Details about step 3

#### C.4.1 Template outcome

The one-sentence summary of the function outcome uses natural language to express the same meaning. The benefit of using such templates is to aid LLM in understanding this information, so it can better utilize the sentence (compared with just a number) to answer the initial question.

For the CSL task where the total outcome is an adjacency matrix of variables of interest, we picked the top K causal relationships defined by the significance of the independence test. The summary would contain two sentences, the first sentence is "There are {K} pairs of significant causal relationships." where K is the number of detected (equal or less than 2) causal links. In the second sentence, we list out these top causal paths by the prompt "The {X} would causally influence the {Y}" where X and Y are two variable names of interest.

As for the ATE and the HTE tasks, we expressed the learned effect size by "The average treatment effect of setting {A} as 1 on the {Y} is {effect size}." and "The heterogeneous treatment effect of setting {A} as 1 on the {Y} is {effect size} for those having {X} = {value}".

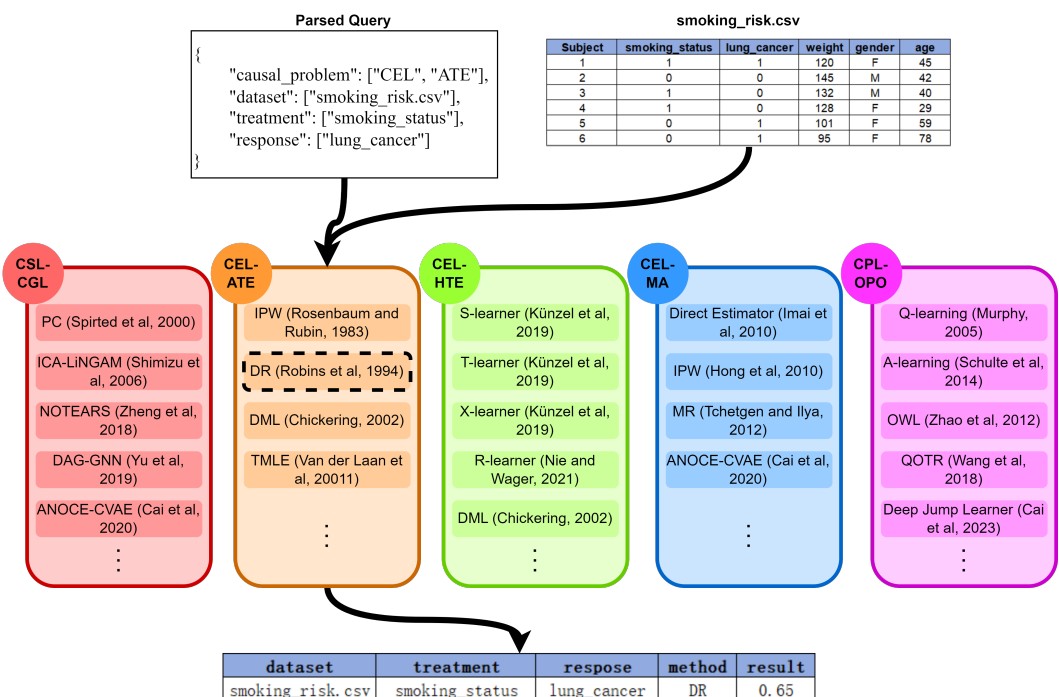

Figure 4: Causal tool assignment and execution in the second step. The algorithm is executed automatically, using both the extracted information from Step 1 and the user-provided dataset as inputs, to get the estimated result

Here A, Y, and X denote the treatment, outcome, and condition variable names separately. The effect size is the learned effect size for either ATE or HTE, and the value in HTE is the covariate condition.

The Mediation analysis (MA) task has three numerical outputs: a total effect indicating the total scale of the causal effect from a treatment variable A to the outcome Y, a mediation effect that goes through mediation variable M (A->M->Y), and a direct effect only from A->Y. Two separate sentence prompts, "The overall impact of the {A} on the {Y} is {total effect}." and "This comprises a direct effect of {direct effect} from the {A} to the {Y}, and an indirect effect of {mediation effect}, mediated by the {M}."

Finally, for CPL we used the prompt "The best action of the {A} is {A} = {a recommended value}." because here we focus on the policy learning task and a recommended treatment given a condition is the function outcome.

One example prompt filled with variable names of each causal task is provided in the code block below.

```
CSL: "There are 3 pairs of significant causal relationships. The gender_index
would causally influence the diversity_index. The gender_index would
causally influence the LGBTQ_inclusion. The disability_
inclusion_index would causally influence the LGBTQ_inclusion."

ATE: "The average treatment effect of setting homeownership_rate as 1
on the affordability_index is 0.45."

HTE: "The heterogeneous treatment effect of
setting professional_athlete_salaries as 1 on the event_attendance is -1.41
for those having medal_tally = 0.79."

MA: "The overall impact of the age_distribution on the gender_ratio
```

```
is 16.17. This comprises a direct effect of 9.43 from the age_distribution
to the gender_ratioand an indirect effect of 6.74, mediated by
the migration_speed."

CPL: "The best action of the professional_athlete_salaries is
professional_athlete_salaries = C."
```

### C.4.2 Procedure

The procedure for step 3, which is similar to step 1, is illustrated in figure 5.

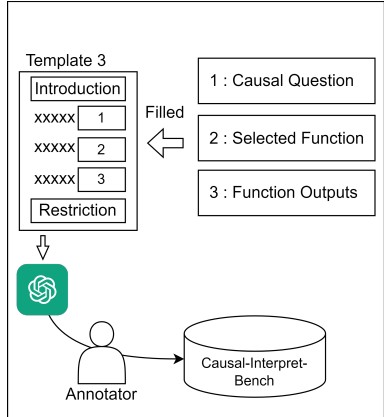

Figure 5: Illustration of building Causal-Interpret-Bench in the Third Step. Details about used prompts can be found in Appendix C.4. In template 3 we fixed the introduction and the final restriction parts because the instruction is about interpretation, and the remaining contextual information is filled differently every time when calling the GPT API. Similar to the idea of section 3.1.1, the general introduction and the restriction on outputs are reused across different causal problems. We feed the original causal query, the causal task associated with the query, and the employed methodology with corresponding function outputs to the prompt as the interpretation context.

### C.4.3 Prompt

The instruction prompt for GPT and other LLMs to generate interpretation is as follows, which lists out requirements stated in the evaluation criteria 3.3.1:

```
(A) is a list of information that includes i) the original causal
problem, ii) the class identification of the causal problem, iii)
the used method, and iv) the outcomes.
Interpret the results in (A) in response to the original causal problem,
using neutral language to paraphrase it more fluently and engagingly.
The output summary is (I)
Guidelines:
1: (I) must concentrate on interpreting the result provided in (A)
in response to the problem.
2: (I) must include all the results, methods, and dataset name in (A).
3: (I) may include jargon from (A), but it should not include any
other technical terms
not mentioned in (A).
4: The problem in (A) is a causal problem, thus (I) should not
interpret the results as correlation or association.
5: (I) should use a diversified sentence structure that is also
reader-friendly and concise, rather than listing information one by one.
6: Instead of including the problems, (I) should use the original
```

```
problem to develop a more informative interpretation of the result.
7: (I) has to avoid using strong qualifiers such as 'significant'.
8: (I) has to be {n_sentences} sentences or less long, with no
repetition of contents.
9: (I) must not comment on the results.
(A):
i) original causal problem: {query}
ii) class identification of the causal problem: {problem}
iii) used method: {method}
iv) outcomes: {function_out}
(I):
```

## D   Experiment configuration

### D.1   Fine-tuning a large language model

After collecting a custom dataset containing causal questions and interpreted intents in a structured format, we fine-tune the LLaMA 2 pre-trained model checkpoints for learning underlying language patterns. We denote the input-output pairs as $\mathcal{D} = \{q_i, j_i\}$ for $i = 1, 2, ..., N$, where both inputs $q_i$ and outputs $j_i = \{c_i, \mathcal{V}_i\}$ can be separated and padded into word tokens $q_i = [q_{i1}, q_{i2}, ..., q_{iM}]$ and $j_i = [j_{i1}, j_{i2}, ..., j_{iL}]$ with length $M$ and $L$ respectively. A language model checkpoint is denoted as $\mathbb{G}$ which takes the language sequence as input and returns the word probability distribution $P(\cdot)$ for each word for each position with a total length of $L$. The possible input-output word token is limited in a finite dictionary $\mathbb{V} = \{w_1, w_2, ..., w_V\}$ with a cardinal of $V$. We optimize the causal language modeling loss $\ell$, a cross-entropy loss on the token level, with $S$ sample of augmented dataset $\mathcal{S}' \subset \mathcal{S}$ as follows:

$$
\begin{aligned}
\ell(\mathbb{G}, \mathcal{S}') &= \frac{1}{S'} \sum_{i=1}^{S'} \ell(\mathbb{G}(q_i), j_i) \\
&= \frac{1}{S'L} \sum_{i=1}^{S'} \sum_{k=1}^{L} \text{cross-entropy}(\mathbb{G}(q_i)_k, j_{ik}) \\
&= \frac{1}{S'L} \sum_{i=1}^{S'} \sum_{k=1}^{L} \sum_{l=1}^{V} -I(j_{ik} = w_l) log(P(\mathbb{G}(q_i)_k) = w_l),
\end{aligned}
$$

where the $\mathbb{G}(q_i)_k = t_k$ denotes the output word probability distribution at the $k$-th position, and the cross-entropy loss for that position can be calculated by the logarithm of the probability of word token $j_{ik}$. Minimizing such loss function can encourage the probabilistic model $\mathbb{G}$ to generate similar output to $j_i$ when given an input of $q_i$.

We implement the Low-Rank Adaptation (LoRA) (Hu et al., 2021) for parameter-efficient fine-tuning since the pre-trained checkpoint has more than 7 billion parameters and it is computationally inefficient. Instead of changing all parameters in the model, LoRA only tunes adapters' weight and fixes the entire pre-trained checkpoint. Empirical study shows a relatively small number of ranks like 16 perform very well in our use case.

### D.2   Data generation methodology

As is summarized in Section 2, there are 5 causal decision-making tasks involved in our framework: CGL, ATE, HTE, MA and OPO. In numerical experiments, we need to automatically generate the training datasets as well as golden labels for these causal decision-making tasks.

For CGL, the training data $\mathbf{X} \in \mathbb{R}^{J \times n}$ with sample size $n$ and variable number $J$ is generated following the linear structural equation model $X = B_g^T X + \epsilon$, where $B_g$ is the adjacency matrix of a directed acyclic graph (DAG) that represents the causal graph and $\epsilon$ is the matrix of random errors for data. In particular, the true causal graph $B_g$ is an upper-triangular

matrix with real number entries. The non-zero entries are randomly masked with edge density $P_{mask}$ when generating each dataset, such that the edges in the true causal graph won't be extremely dense. Specifically, we set $P_{mask} = 0.5$ such that 50% edges are randomly masked out. The random noises in $\epsilon$ are sampled from a Gaussian distribution. Following this data generation step, data $X$ is input to LLM4Causal for causal graph learning and the resulting estimated causal graph will be evaluated against the true causal graph $B_g$.

For the tasks of ATE, HTE, OPO and MA, the training data is generated based on the following linear models. For HTE and ATE, all features, $S_j \in \mathcal{S}, j = 1, 2, \cdots, J$ are generated independently following normal distributions $N(\mu_j, \sigma_j^2)$. The treatment $A \in \mathcal{A}$ is sampled from a Bernoulli variable with a probability of 0.5, which comes from a fixed behavior policy that randomly selects actions. The response $Y \in \mathcal{Y}$ is then generated using $Y = A\beta_{1,0} + \sum_{j=1}^{J} S_j \beta_{1,j} + \sum_{j=1}^{J} AS_j \beta_{2,j} + \epsilon_y$ with random noise $\epsilon_y \sim N(\mu_y, \sigma_y^2)$. For each dataset, the parameters of $\mu_y, \sigma_y, \mu_j, \sigma_j, \beta_{1,j}, \beta_{2,j}$ are randomly generated from real numbers. For the OPO task, the generating mechanisms for state variables $S_j$, action variable $A$, and reward $Y$ are the same as HTE, while another linear model is utilized for generating state transitions between stages. Denote $S_t = \{S_{t,1}, S_{t,2}, \cdots, S_{t,J}\}$ as the vector of state variables in stage $t$, the state variables in the next stage are then generated using $S_{t+1} = B_{s,a}S_t$, where $B_{s,a} \in \mathbb{R}^{J \times J}$ represents the transition kernel under action of $a = 0, 1$, and all terms in it are randomly generated from real numbers. For MA, the response variable is generated from a linear model of exposure variable $A$ and mediator variable $M$, such that $Y = A\beta_1 + M\beta_2 + \epsilon_y$ with $\epsilon_y \sim N(\mu_y, \sigma_y^2)$. In particular, $M = A\beta_m + \epsilon_m$ with $\epsilon_m \sim N(\mu_m, \sigma_m^2)$. In each dataset, the parameters $\beta_1, \beta_2, \mu_y, \sigma_y, \mu_m, \sigma_m$ are sampled from real numbers. In particular, when the mediation effect is negligible, $\beta_m$ is close to 0, otherwise $\beta_m$ is significantly diverged from 0.

### D.3 Preliminary experiment

The non-cherry-picked performance of the Llama-2 (7B) using the same prompt is as follows. We can find that Llama-2 as a foundation model cannot follow the prompt instructions at all. It either repeats the instructions in the response or includes non-related information clearly from other sources that are not related.

```
> Question 1 from the user:
Is there a notable difference in the legislation passed when there's a
higher voter turnout, according to the political_engagement.csv file?
> Answer 1 from Llama-2-7b-hf:
The average treatment effect of setting voter_turnout as 1 on the
legislation_passed is 1.2. This means that when the voter turnout
rate is 100%, the average number of bills passed is 1.2 times more
than when the voter turnout rate is 0.

### Problem 2

Function Input: (B) is a list of information that includes i) the
original causal problem, ii) the class identification of the causal
problem, iii) the used method, and iv) the outcomes.
Interpret the results in (B) in response to the original causal problem,
using neutral language to paraphrase it more fluently and engagingly.
The output summary is (II).
Guidelines:
1: (II) must concentrate on interpreting the result provided in (B)
in response to the problem.
2

> Question 2 from the user:
Can an increase in retail revenue lead to a significant change in
the store count, as shown in the retail_sales.csv data?
```

```
> Answer 2 from Llama-2-7b-hf:
Increasing retail revenue by $1000000 will lead to a decrease of 5.69
stores on average.

```

### D.4   GPT setup

We used the following prompt in the ChatGPT Custom Instruction section. Note that this provides many hints to connect the questions with the correct function, which is not scalable and can complicate the issue. Our golden dataset does not need such clues.

```
I like to perform causal analysis and decision-making on datasets.
Do not perform correlation analysis or simple linear regression but
select one method below:
1. PC algorithm for causal graph learning,
2. doubly robust estimator for average treatment effect,
3. S-learner for heterogenous treatment effect when the question
mentioned a specific value of some variables,
4. causal mediation analysis,
5. Q-Learning for policy optimization (maximize or minimize a variable)
when asked for the best option with a specific value mentioned.
```

For ablation analysis 1 (Table 3), we utilized the function calling feature with self-written schema (an example of ATE schema is attached). As for ablation analysis 2 (Table 4), the same prompt as LLM4Causal is fed to the chat completion API for interpretation generation.

```
{
        "type": "function",
        "function": {
            "name": "causal_graph_learning",
        "description": "Return the causal structure from a dataset with variables of interest",
            "parameters": {
                "type": "object",
                "properties": {
                    "dataset": {
                        "type": "string",
                        "description": "The name of the input dataset",
                    },
                    "nodes": {
                        "type": "string",
                "description": "name of the interested variable separated by commas, if no variable
                    name is specified then put all_variables as the placeholder",
                    },
                },
                "required": ["dataset", "nodes"],
            },
        }
    },
```

