# OpenReview forum: "LLM4Causal: Democratized Causal Tools for Everyone via Large Language Model"
_colmweb.org/COLM/2024/Conference — COLM_

### Official Review · Reviewer_vBxV · 2024-05-07

**Rating:** 8
**Confidence:** 3
**Ethics Flag:** 1

**Summary:**

The authors propose a fully end-to-end pipeline generating responses to user querying an input dataset. The query consists of a causal question which need statistical causal analysis. The goal is for the proposed system to

1) understands the natural-language expressed query to format it in the correct way so that

2) the appropriate inferred tool is run through a predefined list of tools available online whose output is

3) formatted itself in a response phrased in natural language and answering the initial query

The proposed approach, LLM4Causal, describes how the authors fine-tuned a modern pre-trained LLM on two datasets with the aim of improving the LLM output with regard to subtasks 1) and 3). The semi-automated methods to create these datasets are presented in details.

The authors propose automatic and manual evaluation metrics and compare their system to the function calling feature of a chat-GPT (GPT4-turbo) LLM.

Their experimental results demonstrate the superiority of their approach over the benchmark method.

**Questions To Authors:**

Minor comments:
- Table 1: overlapping caption
- 3.1: adopt → adapt?
- 3.1: publicity → publically?
- 3.2: Appendix ??

**Reasons To Accept:**

The method is clearly described, experiments are sound. The task addressed is an effort toward leveraging natural language understanding and generation capabilities of LLM for actual assistant responding accurately to queries in the domain of causal analysis of quantitative variables.

The work appears to be novel, useful for future steps in this direction, well designed and correctly evaluated.

**Reasons To Reject:**

Some details are omitted or not clearly explained, such as how exactly are the tools selected: does LLM4Causal’s LLM handle all the logic or do you implement standard logic once the causality-related task is identified by LLM4Causal?

In other words, once LLM4Causal predicts that the task is CEL-ATE, how is DR selected over DML, TMLE, or IPW?

---

> ### Author Rebuttal · Authors · 2024-05-31
>
> We sincerely appreciate your careful reading and insightful questions, and we are honored to hear that you believe the proposed method is novel and useful. The following are our point-by-point responses.
>
> **Tool selection**. Since the major focus of this work is the LLM’s causal reasoning ability, we only set a default method for each identified task for the end-to-end evaluation experiment, e.g., we assign the doubly robust method for all the identified ATE tasks in experiment 4.1. The main purpose of the experiment is to check whether an LLM has been equipped with causal tool calling and interpretation abilities.
>
> However, we agree that method selection is of great importance for practical use and are currently developing a systematic framework to automatically select the method for each query. Some of the potential future approaches could be building a machine-learning-based classification model for method selection [1,2,3], analyzing data structures to check model assumptions, and interacting with users to allow them to specify their preferred methods. All these potential ways have their own benefits and require time and effort to construct models and compare performance. Thus, we will leave the model selection automation as future research directions and keep the current work focused more on the LLM part.
>
> **Typos and overlapping captions**. We will fix them in the final version. Thank you for pointing this out.
>
> [1] Zhang, J., Jennings, J., Zhang, C., & Ma, C. (2023). Towards causal foundation model: on duality between causal inference and attention. arXiv preprint arXiv:2310.00809.
>
> [2] Gupta, S., Zhang, C., & Hilmkil, A. (2023). Learned Causal Method Prediction. arXiv preprint arXiv:2311.03989.
>
> [3] Lorch, L., Sussex, S., Rothfuss, J., Krause, A., & Schölkopf, B. (2022). Amortized inference for causal structure learning. Advances in Neural Information Processing Systems, 35, 13104-13118.

---

> > ### Comment · Reviewer_vBxV · 2024-06-04
> > **Thank you for the clarification**
> >
> > Thank you for the clarification.
> >
> > I now understand that, after identifying the task, the method is selected by default. In my opinion (also raised by Reviewer UqM4), it would improve readability if you specified that each task is currently assigned a default method, and future steps could include automatic method selection for a given detected task (out of this work's scope).

---

> > > ### Author Response · Authors · 2024-06-05
> > > **Response**
> > >
> > > Dear reviewer, thanks very much for your detailed comments and valuable suggestions. We will update them in our revised paper.

---

### Official Review · Reviewer_UqM4 · 2024-05-12

**Rating:** 4
**Confidence:** 4
**Ethics Flag:** 1

**Summary:**

This work explores the capability of large language models(LLMs) to perform causal tasks. The authors started with the motivation that existing LLMs do not handle specific structured data and knowledge well. The objective of this work is to build LLM4Causal by fine-tuning existing open-source LLMs capable of identifying causal tasks executing the corresponding functions and outputting the results. To fine-tune the training, they constructed two supervised datasets with GPT-4 prompting.

**Questions To Authors:**

The appendix provides insufficient details on how to use the GP4-turbo.

**Reasons To Accept:**

This work proposes a complete pipeline for boosting the performance of LLMs on specific tasks. However, constructing specific datasets via GPT-4 prompts and then fine-tuning open-source LLMs has been used in many previous works.

**Reasons To Reject:**

1. The work did not really improve LLMs' abilities in causal decision-making and reasoning, but rather taught LLMs to use causal analysis tools.
2. The experiment was well-designed and predictable.
3. Larger language models with larger scale parameters generally have stronger reasoning capabilities. It is suggested that further exploration and experimentation be conducted.
4. Provide implications in calling different causal tools and algorithms.

---

> ### Author Rebuttal · Authors · 2024-05-31
>
> Thanks for the insightful comments, please let us know if we misunderstood any of them.
>
> **LLM to use tools**. Our decision not to use LLMs’ inherent abilities for causal decision-making is due to their reliance on knowledge acquired during pre-training. Without such knowledge, LLMs' numerical analysis capabilities on causal tasks are rather limited [1].  When data for causal inference conflicts with LLMs’ stored knowledge, this intrinsic ability can be inadequate. Moreover, the integration of external tools into LLM frameworks is an actively expanding research area, which motivates our approach. The complexities and challenges of this integration, discussed below, highlight the novelty of our work.
>
> **Experiment design**. First, identifying the correct type of causal question in STEP1 is challenging for both LLMs and humans. Our model excels in STEP1 due to a clearly defined JSON format for causal problems and fine-tuning on our proposed Causal-Retrieval-Bench data. Similarly, interpreting external tools' results in STEP3 is also non-trivial, as even advanced LLMs like GPT-4 can confuse causation with correlation. The satisfactory results in Table 4 are due to our carefully designed interpretation evaluation metrics in Section 4.2 and the Causal-Interpret-Bench data. Thus, our model's achievements in Tables 2, 3, and 4 are non-trivial and not easily predictable.
>
> **Parameter size**. Indeed, larger LLMs may offer improved capabilities. It is noted that our framework and proposed benchmarks are adaptable to LLMs of any size, not just the Llama-2 7B used in our studies. We selected the 7B model as it was a common choice of prior works and already provided satisfactory accuracy while minimizing computational resources.
>
> **Select causal tools**. In the experiments, a default causal analysis tool is assigned in step 2 for each causal task to maintain simplicity. While choosing the optimal causal analysis tool can be beneficial, it is beyond our scope and left for future research.
>
> **GPT4-turbo usage**. For ablation analysis 1 (Table 3), we utilized the function calling feature with self-written schemas (will be provided in the appendix). As for ablation analysis 2 (Table 4), the same prompt as LLM4Causal is fed to the chat completion API for interpretation generation.
>
> [1] Cai, H., Liu, S., & Song, R. (2023). Is Knowledge All Large Language Models Needed for Causal Reasoning?. arXiv preprint arXiv:2401.00139.

---

> > ### Author Response · Authors · 2024-06-05
> > **Follow-Up**
> >
> > Dear Reviewer, we sincerely appreciate the time and effort you invested in evaluating our submission. We hope our responses have adequately addressed your question and clarified any concerns regarding our paper. Should you have any further feedback, questions, or comments, please do not hesitate to let us know before the rebuttal period concludes. Your insights are invaluable, and we're keen to address any remaining issues.

---

> > ### Comment · Reviewer_UqM4 · 2024-06-06
> >
> > Thanks for the response from the authors. I still have some concerns. External causal tools need to be used under assumptions. How does LLM4Causal address this?

---

> > > ### Author Response · Authors · 2024-06-07
> > > **Response**
> > >
> > > Dear reviewer,
> > >
> > > Thank you for your insightful question! Currently, LLM4Causal can help with some assumption checking. For example, in the current framework, users could request LLM4Causal to validate structure-wise assumptions by asking a causal discovery query. The learned structure among variables can then be used to detect the presence of instrument/mediator/confounder variables, which is critical for accurately performing causal effect estimation tasks. However, other assumptions, such as Markov properties [1], error distributions [2], no unmeasured confounders [3], and so on, remain challenging to verify using observational datasets. This is still an open question in the literature, and hence beyond the scope of what language models can handle.
> > >
> > > We strongly agree that assumption-checking is essential for causal learning.  Therefore, we would explore pre-trained ML models for assumption classification [4] as our next step. Also, it is worth noting that the LLM4Causal framework can easily adapt any assumption-checking tools that might be developed in the future, to improve its capabilities.
> > >
> > > Please do not hesitate to contact us if you have any further questions or require additional clarification. Should we have addressed your concerns, we would appreciate it if you could consider reevaluating the score. Thank you for considering our responses.
> > >
> > > [1] Sharma, A., Syrgkanis, V., Zhang, C., & Kıcıman, E. (2021). Dowhy: Addressing challenges in expressing and validating causal assumptions. arXiv preprint arXiv:2108.13518.
> > >
> > > [2] Celli, V. (2022). Causal mediation analysis in economics: Objectives, assumptions, models. Journal of Economic Surveys, 36(1), 214-234.
> > >
> > > [3] Imbens, G. W., & Rubin, D. B. (2015). Causal inference in statistics, social, and biomedical sciences. Cambridge university press.
> > >
> > > [4] Gupta, S., Zhang, C., & Hilmkil, A. (2023). Learned Causal Method Prediction. arXiv preprint arXiv:2311.03989.

---

### Official Review · Reviewer_y23N · 2024-05-12

**Rating:** 7
**Confidence:** 4
**Ethics Flag:** 1

**Summary:**

This paper presents an end-to-end framework for causal decision-making tasks, which addresses the weakness of the current LLM applications on causal tasks.

A three-step data generation pipeline is proposed, it first interprets user requests into a JSON data for causal task , and then automatically selects the causal tool to execute the corresponding algorithm, finally it generates a natural language output.

Two  benchmark datasets, Causal- Retrieval-Bench for causal function calling and Causal-Interpret-Bench for causal interpretation are constructed.

**Reasons To Accept:**

The article proposes a scheme to transfer diversity causal tasks into a unified structured JSON data. This approach can effectively integrates tasks and facilitating the use of LLM.
The experimental results shows that the method has significant advantages.
The paper provides detailed attachments to demonstrate the effectiveness of the work.

**Reasons To Reject:**

The paper proposes the use of structured JSON to represent a variety of tasks and demonstrates that this integrated approach outperforms end2end GPT-4 method. However, the paper does not validate whether this method is superior to handling each task individually, particularly in cases of individual fine-tuning them. Moreover, controlling the quality of JSON data needs additional manual consumption, which diminishes the practical significance of this integrated approach.

---

> ### Author Rebuttal · Authors · 2024-05-31
>
> We sincerely appreciate your careful reading and insightful comments, and we are honored to hear that you believe the proposed method is effective. The following are our point-by-point responses to your comments.
>
> **Individual task handling**. Thank you for the constructive suggestion. While it is possible to handle each task individually by fine-tuning separate LLM frameworks, our approach is driven by the needs of a general audience who may lack a background in causal analysis. Determining the specific causal task they aim to perform is a significant challenge for these users, and thus a core component of our proposed framework is using a fine-tuned LLM to understand the user's intention and classify their query into a specific causal task. This classification step then helps identify the appropriate causal functions to be used. Therefore, we believe that fine-tuning for each causal task individually is not suitable. We will include additional discussion in the final version to highlight the importance and motivation for integrating various causal tasks.
>
> **Manual consumption**. Thank you for another valuable comment. First of all, it is important to note that users can directly use our fine-tuned LLM4Causal for causal analysis without any additional manual effort. Also, for researchers looking to fine-tune other base LLMs, our two well-prepared instruction-tuning datasets can be used directly to enhance any other user-selected base LLM's capability in causal task classification and interpretation, without the need for additional human annotation. Moreover, we agree that manual efforts to prepare a golden dataset for fine-tuning are unavoidable for ensuring dataset quality, as is common in existing literature (e.g., [1,2]). To minimize the manual efforts needed for preparing the JSON data, we proposed an output-first strategy (detailed in Section 3.1.1). This strategy effectively ensures the diversity and accuracy of the generated JSON and significantly reduces the amount of human evaluation required, limiting it to evaluating the corresponding GPT-4-generated causal queries for the Causal-Retrieval-Bench dataset.
>
> [1] Zhang, J. (2023). Graph-toolformer: To empower llms with graph reasoning ability via prompt augmented by chatgpt. arXiv preprint arXiv:2304.11116.
>
> [2] Qin, Y., Liang, S., Ye, Y., Zhu, K., Yan, L., Lu, Y., ... & Sun, M. (2023). Toolllm: Facilitating large language models to master 16000+ real-world apis. arXiv preprint arXiv:2307.16789.

---

> > ### Author Response · Authors · 2024-06-05
> > **Follow-Up**
> >
> > Dear Reviewer, we sincerely appreciate the time and effort you invested in evaluating our submission. We hope our responses have adequately addressed your question and clarified any concerns regarding our paper. Should you have any further feedback, questions, or comments, please do not hesitate to let us know before the rebuttal period concludes. Your insights are invaluable, and we're keen to address any remaining issues.

---

### Official Review · Reviewer_qRr6 · 2024-05-17

**Rating:** 7
**Confidence:** 4
**Ethics Flag:** 1

**Summary:**

The paper presents LLM4Causal, an end-to-end, user-friendly large language model designed for causal decision-making tasks. The model addresses the limitations of general LLMs in handling causal tasks by fine-tuning them with two specialized datasets: Causal-Retrieval-Bench for identifying causal problems and extracting variables and Causal-Interpret-Bench for interpreting causal results. LLM4Causal operates through three major steps: interpreting user requests, assigning and executing causal tools, and providing easy-to-understand interpretations of the outputs. The model's performance was evaluated across three main causal decision-making categories: causal effect estimation, causal structure discovery, and causal policy learning. In total, there are 5 tasks in these three categories. The results demonstrate superior performance compared to baselines like ChatGPT and GPT-4. This work highlights the potential of fine-tuning large language models to enhance their capabilities in specialized domains like causal inference, making them accessible and effective for general users.

**Questions To Authors:**

No

**Reasons To Accept:**

1. This work presents a very comprehensive framework for causal inference. Here, the authors fine-tuned LLM to interpret uses' causal query, utilize external tools to calculate causal relations, and interpret the logic behind causal relations.
2. The framework covers five tasks and conducts a lot of experiments, showing its generalization ability on various causal tasks.
3. The problem studied in this paper is significant, which is how to use LLM to make causal inference more accessible to normal people.

**Reasons To Reject:**

1. There are some typos, like in Section 3.2, "See Figure 5 in Appendix ??"
2. Since the paper just fine-tunes Llama2 on those causal tasks instead of instruction tuning. Can we just fine-tune a small model, like T5, BART, or GPT2, to do this task?
3. Two fine-tuning datasets collected in this paper are quite small. The Causal-Retrieval-Bench only contains 1,500 pairs. Causal-Interpret-Bench only contains 400 examples. It seems that there will a very sever over-fitting problem for Llama2 (7B).
4. Is this framework possible to be extended to causality between textual events, like the following references?
5. There are also some missing reference:
1) Rashkin, Hannah, et al. "Event2Mind: Commonsense Inference on Events, Intents, and Reactions." Proceedings of the 56th Annual Meeting of the Association for Computational Linguistics (Volume 1: Long Papers). 2018.
2) Ning, Qiang, et al. "Joint Reasoning for Temporal and Causal Relations." Proceedings of the 56th Annual Meeting of the Association for Computational Linguistics (Volume 1: Long Papers). 2018.
3) Wang, Zhaowei, et al. "COLA: Contextualized Commonsense Causal Reasoning from the Causal Inference Perspective." Proceedings of the 61st Annual Meeting of the Association for Computational Linguistics (Volume 1: Long Papers). 2023.

---

> ### Author Rebuttal · Authors · 2024-05-31
>
> Thanks for the insightful comments and careful reading. We will fix the typos in the final version. Our point-by-point responses to your comments are below:
>
> **Model Choices**. We have conducted additional experiments using T5-base(220M) and T5-large model(770M) for step 1, under the same setting as we used in the paper. The table below summarizes the accuracy of task classification and entity extraction, showing that smaller models cannot acquire the necessary abilities for our use case.
>
> ||Causal Task|Dataset|Nodes|Treatment|Response|Mediator|Condition|
> |---|---|---|---|---|---|---|---|
> |t5-base|0.00|0.07|0.13|0.10|0.12|0.03|0.15|
> |t5-large|0.44|0.45|0.47|0.44|0.47|0.07|0.73|
> |LLM4Causal-Mixed|0.98|1.00|1.00|0.96|0.97|1.00|1.00|
>
> As shown in empirical results, smaller models are inadequate for our use case, due to larger model’s emergent abilities that may greatly improve the performance [1]. Meanwhile, using a 7B model with fine-tuning is a common practice based on our reference papers [Schick et al., 2023; Qin et al., 2023b; Zhang, 2023].
>
> **Overfitting**. We want to emphasize that the testing data used in our experiments have varied sentence structures and cover different multi-discipline topics when comparing with the datasets prepared for fine-tuning, as explicitly stated in Section 4. We carefully designed the data generation process (detailed in Section 3.1.1) and implemented prompting techniques such as random demonstration sampling to make sentences further differ from each other. Finally, we want to point out that the scale of our dataset is at the same magnitude as [Qin et al., 2023b and Zhang, 2023] and there are no information leaks between training and inference procedures.
>
> **Causality between textual events**. LLMs’ causal ability from text events, similar to prompting LLMs on causal link existence, focuses on common sense induction [Rashkin et al., 2018; Wang et al., 2023]. Such internal knowledge is inadequate if the pattern does not match newly arrived data, which motivates us to enhance LLMs' causal ability via external tools. Merging internal (from common sense) and external (from data or specific conditions) causal analysis results would be a valuable future direction. We will include more discussion about the related references in the final version.
>
> [1] Wei, J., Tay, Y., Bommasani, R., Raffel, C., Zoph, B., Borgeaud, S., ... & Fedus, W. (2022). Emergent abilities of large language models. arXiv preprint arXiv:2206.07682.

---

> > ### Author Response · Authors · 2024-06-05
> > **Follow-Up**
> >
> > Dear Reviewer, we sincerely appreciate the time and effort you invested in evaluating our submission. We hope our responses have adequately addressed your question and clarified any concerns regarding our paper. Should you have any further feedback, questions, or comments, please do not hesitate to let us know before the rebuttal period concludes. Your insights are invaluable, and we're keen to address any remaining issues.

---

> > ### Comment · Reviewer_qRr6 · 2024-06-05
> > **Response to Rebuttal**
> >
> > I have read the response

---

### Decision · Program_Chairs · 2024-07-10

**Decision:**

Accept

**Comment:**

This paper makes a useful contribution by using LLMs as an interface to using tools for causal decision making. The idea is neat and well-executed. I agree that the paper should clarify that the LLMs are not used for causal decision making, but to interface to appropriate tools, as the title at the moment is misleading (reviewer UqM4), but I still think the paper has merit.

[comment from the PCs] Please follow the AC recommendation to revise the title.